 SciPost Phys. Lect. Notes 42 (2022)

# Dark Matter Superfluidity

**Justin Khoury**[⋆]

Center for Particle Cosmology, Department of Physics and Astronomy,
University of Pennsylvania, Philadelphia, PA 19104

⋆ jkhoury@upenn.edu

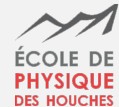

*Part of the Dark Matter*
*Session 118 of the Les Houches School, July 2021*
*published in the Les Houches Lecture Notes Series*

## Abstract

In these lectures I describe a theory of dark matter superfluidity developed in the last few years. The dark matter particles are axion-like, with masses of order eV. They Bose-Einstein condense into a superfluid phase in the central regions of galaxy halos. The superfluid phonon excitations in turn couple to baryons and mediate a long-range force (beyond Newtonian gravity). For a suitable choice of the superfluid equation of state, this force reproduces the various galactic scaling relations embodied in Milgrom's law. Thus the dark matter and modified gravity phenomena represent different phases of a single underlying substance, unified through the rich and well-studied physics of superfluidity.

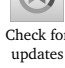

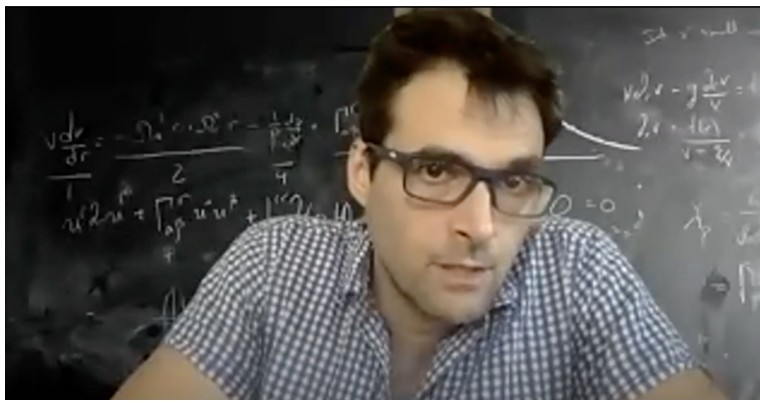

## 1   Introduction

Dark matter (DM) is by now established as a pillar of modern cosmology, yet its fundamental nature remains unknown. One thing we know about DM is that on the largest scales it must behave as a cold, collisionless fluid. This provides an exquisite fit to a host of cosmological observations, from the expansion history to the cosmic microwave background (CMB) temperature anisotropies to the rate of growth of cosmic structuress.

The standard $\Lambda$ Cold Dark Matter ($\Lambda$CDM) model assumes the simplest form of DM — a single species of (effectively) collisionless particles, such as weakly interacting massive particles (WIMPs) or axions. On the other hand, large-scale observations only probe the hydrodynamical limit of DM; any perfect fluid with sufficiently small pressure and sound speed would do equally well at fitting the data on those scales. There is in principle room for new physics on non-linear scales, particularly in galaxies.

Indeed, while the $\Lambda$CDM model has been remarkably successful at matching cosmological observations on large scales, its scorecard on galactic scales has been the subject of active debate [1, 2]. Despite the complex role of baryonic feedback processes in their formation, galaxies are surprisingly regular, exhibiting striking correlations among their physical properties. Disc galaxies display a remarkably tight correlation between the total baryonic mass (stellar + gas) and the gravitational acceleration in galaxies [3–5]. This apparent conspiracy, known as the radial acceleration relation (RAR), states that the acceleration experienced by a baryonic particle (irrespective of whether it is due to DM, modified gravity, or both) can be *uniquely* predicted from the baryon density profile. At large distances, the RAR implies the baryonic Tully-Fisher relation (BTFR) [6], which relates the total baryonic mass to the asymptotic/flat rotation velocity as $M_{\mathrm{b}} \sim V_{\mathrm{f}}^4$. This relation holds over five decades in mass, with remarkably small scatter. Another scaling relation is the correlation between the central

stellar and dynamical surface densities in disk galaxies [7].

The RAR was predicted nearly forty year ago by Milgrom [8] with MOdified Newtonian Dynamics (MOND). The MOND law states that the gravitational acceleration $a$ is related to the baryonic acceleration $a_b$ (*i.e.*, the Newtonian acceleration due to ordinary matter alone) via

$$a = \begin{cases} a_b & a_b \gg a_0 \\ \sqrt{a_b a_0} & a_b \ll a_0 \, , \end{cases} \tag{1}$$

where $a_0$ is a characteristic acceleration scale. Its best-fit value is intriguingly of order the speed of light times the Hubble constant $H_0$:

$$a_0 \simeq \frac{1}{6} c H_0 \simeq 1.2 \times 10^{-8} \text{ cm/s}^2 \, . \tag{2}$$

All of the aforementioned galactic scaling relations are an exact consequence of this law. MOND does exquisitely well at fitting detailed galactic rotation curves [9–11]. Although originally promoted as an alternative to DM, as an empirical fact the success of the MOND law at fitting galactic properties is unequivocal, especially in rotationally supported systems. The acceleration scale $a_0$ is in the data. Even if DM is made of the most standard, run-of-the-mill WIMPs, its density profile in galaxies must at the end of the day conform to MOND.

At this juncture one can take one of three attitudes:

- The conservative viewpoint is that the MOND law (1) is an emergent phenomenon, due to complex baryonic feedback processes (star formation, supernovae, gas cooling/heating, *etc.*). In other words, DM consists of collisionless particles, it interacts with baryons only through gravity, and the observed properties of galaxies emerge from the interplay of gravity and baryonic feedback effects.

  This possibility faces some challenges. First, semi-empirical models with cored DM halos can reproduce the observed slope and normalization of the BTFR, but not yet the small scatter [12–14]. Second, the diversity of shapes of rotation curves at a given maximal circular velocity remains a puzzle in $\Lambda$CDM [15] (though see also [16]). Lastly, the observation that the central slope of the rotation curve correlates with the baryonic surface density means that feedback processes should be more efficient at removing DM from the central regions of galaxies with lower surface density, which are often more gas-dominated. Hence, feedback efficiency should increase with decreasing star formation rate [6] and be tightly anti-correlated with baryonic surface density (or correlated with scale-length at a given mass scale).

- At the other end of the spectrum is the viewpoint that the MOND law represents a fundamental modification of gravity (*e.g.,* [17,18]). In other words, DM does not exist, and (1) represents a fundamental modification of gravity. This possibility seems unlikely, given the observational evidence for DM behaving as a collisionless fluid on cosmological and cluster scales. Modified gravity theories cannot explain the angular power spectrum of the CMB without additional DM or unrealistically massive ordinary neutrinos [19]. Furthermore, Milgrom's law does not work on galaxy cluster scales, *e.g.*, [20,21].

  Nevertheless, it is instructive to see how (1) can arise from a modification of gravity. The simplest possibility is to postulate the existence of a scalar field, akin to the gravitational potential, with non-relativistic action [17]

$$\mathcal{L} = -\frac{1}{12\pi G_N a_0} \left( (\vec{\nabla}\phi)^2 \right)^{3/2} + \phi \rho_b \, , \tag{3}$$

where $\rho_{\rm b}$ is the baryonic mass density. The equation of motion is a non-linear Poisson equation:

$$\vec{\nabla} \cdot \left( \frac{\left|\vec{\nabla}\phi\right|}{a_0} \vec{\nabla}\phi \right) = 4\pi G_{\rm N} \rho_{\rm b}. \qquad (4)$$

Ignoring a homogeneous curl term (which vanishes, in particular, for spherical symmetry), the solution is $a_\phi = \left|\vec{\nabla}\phi\right| = \sqrt{a_{\rm b} a_0}$. The total acceleration, $a = a_{\rm b} + a_\phi$, is consistent with (1). However, as a theory of a fundamental scalar field, the non-analytic form of the kinetic term is somewhat unpalatable.

- A middle-ground interpretation is that the MOND law is telling us something about the *fundamental nature* of DM. In other words, DM certainly exists, and behaves as a cold, collisionless fluid on large scales. However, the MOND law informs us about the microphysics of DM and its interactions with baryons.

  A number of DM-MOND hybrid theories have been put forth over the years, *e.g.*, [22–31], but such proposals generally face two important challenges. First, there is the potential drawback of having two *a priori* unrelated ingredients: a DM-like component and a modified-gravity component. Second, the theory must be adjusted such as to avoid the phenomenological co-existence of DM-like and MOND-like behavior. The DM component must dominate on large, extra-galactic scales, with negligible modified-gravity contribution, whereas the modified-gravity component must dominate in central regions of galaxies where rotation curves are measured, with negligible DM contribution. How this can be arranged is highly non-trivial. The superfluid approach described here overcomes both challenges.

These lectures describe a unified framework for the DM and MOND phenomena, based on DM superfluidity, which brings together concepts of condensed matter physics, cold atom physics and astrophysics [32–35]. In this novel approach, the DM and MOND components represent different phases of a single underlying substance, unified through the rich and well-studied physics of superfluidity. The MOND empirical law emerges from the superfluid phase of DM.

As in $\Lambda$CDM, the model assumes DM particles, which behave as a cold, collisionless fluid on cosmological scales. As non-linear structures form, the increase in DM density triggers a phase transition, causing DM to condense into a superfluid phase. As we will see, this requires DM particles to be sufficiently light, $m \lesssim$ few eV, such that their de Broglie wavelengths overlap, and have self-interactions.

The superfluid nature of DM dramatically changes its macroscopic properties in galaxies. Instead of behaving as individual collisionless particles, the DM is more aptly described as collective excitations, which at low energy/momentum are phonons. Phonons play a key role by coupling to ordinary matter, and thereby mediate an additional force (beyond Newtonian gravity) between baryons. For a particular choice of the superfluid equation of state, the DM superfluid reproduces the MOND law in galaxies.

The possibility of Bose-Einstein condensation (BEC) has been studied before in the context of DM, *e.g.*, [36–48]. A key difference is that in BEC DM galactic dynamics are caused by the condensate density profile, similar to what happens in CDM, with phonons being irrelevant. In the present framework, on the other hand, phonons play a key role in generating the MOND law. For a nice review of ultra-light DM, including DM BEC and superfluidity, see [49].

## 2 Bose-Einstein condensation

Let us begin by briefly reviewing some elementary facts about BEC.[1] Consider a free Bose gas, in chemical and thermal equilibrium with a reservoir, with chemical potential $\mu$ and temperature $T$. Its statistics are described by the grand canonical ensemble.

The probability $p_i$ that the system has $N_i$ particles with single-particle energy $\epsilon_i$ is

$$p_i(N_i) = \frac{1}{Z} e^{\beta N_i (\mu - \epsilon_i)}; \qquad \beta \equiv \frac{1}{k_B T}, \tag{5}$$

where $Z \equiv \sum_i e^{\beta N_i (\mu - \epsilon_i)}$ is the partition function. The average occupation number in state $i$, given by $\langle N_i \rangle = \sum_{N_i=0}^{\infty} N_i p_i(N_i)$, is a geometric series which converges for $\mu < \epsilon_i$. The result is the celebrated Bose-Einstein distribution,

$$\langle N_i \rangle = \frac{1}{e^{\beta(\epsilon_i - \mu)} - 1}. \tag{6}$$

Summing over $i$ gives the total number of particles,

$$
\begin{aligned}
N &= \sum_i \langle N_i \rangle \\
&= \frac{1}{e^{-\beta \mu} - 1} + \sum_{i \neq 0} \frac{1}{e^{\beta(\epsilon_i - \mu)} - 1} \\
&\equiv \quad N_0 \quad + \quad N_{\text{exc}},
\end{aligned}
\tag{7}
$$

where we have split the sum into the ground state occupation number $N_0$, assuming $\epsilon_0 = 0$ without loss of generality, and the occupation number for the excited states $N_{\text{exc}}$. In terms of the *fugacity* $z \equiv e^{\beta \mu}$, the ground state particle number takes the simple form

$$N_0 = \frac{z}{1-z}. \tag{8}$$

In the case of the ground state, convergence of the geometric series requires $-\infty < \mu < 0$, therefore $0 < z < 1$.

Our next task is to simplify the excited state occupation number $N_{\text{exc}}$. To do so, take the continuum limit, $\epsilon_i \to \epsilon \equiv \frac{\vec{k}^2}{2m}$, such that the sum over $i$ becomes an integral over momenta:

$$N_{\text{exc}} = V \int \frac{d^3 k}{(2\pi)^3} \frac{1}{e^{\beta(\epsilon - \mu)} - 1} = V \frac{m^{3/2}}{\sqrt{2}\pi^2} \int_0^{\infty} d\epsilon \frac{\epsilon^{1/2}}{e^{\beta(\epsilon - \mu)} - 1}, \tag{9}$$

where $V$ is the volume of the system, and where the last step follows from doing the angular integral. Letting $x \equiv \beta \epsilon$, the integral becomes

$$N_{\text{exc}} = V \frac{m^{3/2}}{\sqrt{2}\pi^2 \beta^{3/2}} \int_0^{\infty} dx \frac{x^{1/2}}{z^{-1} e^x - 1} = \frac{V}{\lambda_{\text{th}}^3} g_{3/2}(z), \tag{10}$$

where $\lambda_{\text{th}} \equiv \sqrt{\frac{2\pi}{m k_B T}}$ is the thermal de Broglie wavelength, and $g_\nu(z)$ is the polylogarithm function,

$$g_\nu(z) = \frac{1}{\Gamma(\nu)} \int_0^{\infty} dx \frac{x^{\nu-1}}{z^{-1} e^x - 1} = z + \frac{z^2}{2^\nu} + \frac{z^3}{3^\nu} + \dots \tag{11}$$

---

[1] Throughout these lectures, we work in natural units, with $\hbar = c = 1$.

In particular, since $z < 1$, we have $g_{3/2}(z) < g_{3/2}(1) = \zeta\left(\frac{3}{2}\right) \simeq 2.61$, hence the number density $n_{\mathrm{exc}} = N_{\mathrm{exc}}/V$ in excited states is bounded above:

$$n_{\mathrm{exc}} = \frac{g_{3/2}(z)}{\lambda_{\mathrm{th}}^3} < \frac{\zeta\left(\frac{3}{2}\right)}{\lambda_{\mathrm{th}}^3} \,. \tag{12}$$

Now, suppose that more and more particles are added to the system at fixed $T$, such that the total number density $n = \frac{N}{V}$ exceeds the maximal allowed density (12) in the excited states,

$$n > \frac{\zeta\left(\frac{3}{2}\right)}{\lambda_{\mathrm{th}}^3} \,. \tag{13}$$

It follows that the excess particles must inevitably populate the ground state. This macroscopic occupation of the ground state is known as *Bose-Einstein condensation*. At fixed $T$, it occurs at the critical density

$$n_{\mathrm{c}} = \frac{\zeta\left(\frac{3}{2}\right)}{\lambda_{\mathrm{th}}^3} = \zeta\left(\frac{3}{2}\right)\left(\frac{mk_{\mathrm{B}}T}{2\pi}\right)^{3/2} \,. \tag{14}$$

Equivalently, at fixed density, condensation occurs at the critical temperature

$$T_{\mathrm{c}} = \frac{2\pi}{mk_{\mathrm{B}}}\left(\frac{n}{\zeta\left(\frac{3}{2}\right)}\right)^{2/3} \,. \tag{15}$$

## 2.1 Some intuition

At first sight, BEC may not seem particularly surprising, as you intuitively expect some particles to occupy the ground state at sufficiently low temperature. To see that BEC is not simple energetics, as this argument suggests, but rather is a startling consequence of quantum mechanics, let us compare $T_{\mathrm{c}}$ with the temperature $T_{\mathrm{gap}} = \frac{\epsilon_1}{k_{\mathrm{B}}}$ set by the energy of the lowest excited state (recall that $\epsilon_0 = 0$). Since the system is in box, we have $\epsilon_1 = \frac{\vec{k}_{\mathrm{min}}^2}{2m} \sim \frac{1}{mV^{2/3}}$, thus

$$T_{\mathrm{gap}} \sim \frac{1}{mk_{\mathrm{B}}V^{2/3}} \,. \tag{16}$$

In contrast, the critical temperature for BEC, $T_{\mathrm{c}} \sim \frac{N^{2/3}}{mk_{\mathrm{B}}V^{2/3}}$, is larger by a large factor of $N^{2/3}$. In particular, in the thermodynamic limit ($N \to \infty$, $V \to \infty$, with $n = N/V$ fixed), $T_{\mathrm{gap}}$ goes to zero while $T_{\mathrm{c}}$ remains fixed. Thus BEC is not simple energetics — it is a genuine phase transition owing to the quantum mechanics of bosons. Another useful perspective is to consider the fraction of particles in the ground state as a function of temperature. Combining the above results, it is straightforward to show that, for $T \leq T_{\mathrm{c}}$,

$$\frac{N_0}{N} = 1 - \left(\frac{T}{T_{\mathrm{c}}}\right)^{3/2} \,. \tag{17}$$

Thus, as soon as $T$ drops below $T_{\mathrm{c}}$, an order one fraction of particles condense to the ground state.[2]

The criterion (13) has a nice physical interpretation. Ignoring the numerical prefactor, it says that BEC occurs whenever $n \gtrsim \lambda_{\mathrm{th}}^{-3}$, or, equivalently,

$$\lambda_{\mathrm{th}} \gtrsim n^{-1/3} = \ell \,, \tag{18}$$

---

[2]The 3/2 power in (17) is specific to the free Bose gas, and will be different in the presence of interactions and/or external potential. For cold atoms in a harmonic trap, for instance, the power is instead 3 [50].

where $\ell$ is the characteristic inter-particle separation. In other words, the condition for BEC to occur is that the thermal de Broglie wavelengths of particles overlap, which is indeed when quantum mechanics is expected to play an important role.

## 2.2 Dark matter BEC

Let's use the above results to determine the conditions under which DM can form a BEC inside galaxies. The first condition is of course that DM should be effectively bosons, either fundamental bosonic particles or bound states of fermions (*e.g.*, weakly-coupled Cooper pairs or tightly-bound atoms). The second condition is that their thermal de Broglie wavelength in galaxies overlap. Although DM particles are subject to the gravitational field of the galaxy and possibly have self-interactions, we can nevertheless use (18) as a rule of thumb.

As usual, ascribing a DM temperature $k_B T = m v^2$, where $v$ is the one-dimensional velocity dispersion, we have $\lambda_{\text{th}} \sim \frac{1}{mv}$. Thus the condition for BEC is $\frac{1}{mv} \gtrsim n^{-1/3} = \left(\frac{m}{\rho}\right)^{1/3}$, where $\rho = mn$ is the mass density. Rearranging gives an upper bound on the DM mass:

$$m \lesssim \left(\frac{\rho}{v^3}\right)^{1/4}. \tag{19}$$

To fix ideas, let's substitute the matter density and velocity dispersion in our neighborhood of the Milky Way, of order $\rho \simeq 10^{-25}$ g/cm$^3$ $\simeq 4 \times 10^{-7}$ eV$^4$ and $v \simeq 100$ km/s, respectively. Then (19) gives $m \lesssim 6$ eV. A more careful analysis [33] gives $m \lesssim 3$ eV for Milky Way-like galaxies. The key point is that BEC requires DM to be lighter than a few eV. Thus, in the broader spectrum of allowed DM particle masses, BEC DM overlaps with the mass range of axion-like particles.

We can also readily estimate the critical temperature for BEC DM. Substituting the same mass density as before, and taking $m =$ eV for concreteness, (15) gives

$$T_c \simeq 2 \, \text{K}. \tag{20}$$

A more careful analysis [33] gives $T_c \simeq 0.2$ mK. Amusingly, this is not far from the critical temperature of trapped cold atoms, in the $\mu$K range.

## 3 Superfluidity: simplest example

Next we turn to superfluidity, another striking manifestation of quantum mechanics, whereby a system of particles exhibit dissipation-less flow below a critical velocity. Superfluidity and BEC are intimately related phenomena [51–53]. While BEC is necessary for superfluidity, the converse is of course not true — in the absence of interactions, superfluidity disappears, as we will see, whereas BEC can persist without interactions, as emphatically illustrated by the free Bose gas. We will be primarily interested in the case of weakly-interacting bosons, where BEC and superfluidity go hand-in-hand. But it is worth noting that in strongly-interacting systems only a fraction of particles may form a BEC, even at zero temperature. In liquid helium, for instance, only $\sim 10\%$ of particles are in the condensate for $T \ll T_c$, whereas the entire system exhibits superfluidity.

In these lectures, we will describe superfluidity in the language of quantum field theory (QFT), as opposed to, *e.g.*, hydrodynamics, working exclusively at $T = 0$ for simplicity. We will start with a relativistic theory and later take the non-relativistic limit relevant for DM superfluidity. This is done for pedagogical purposes, since students tend to be more familiar with relativistic QFT.

Superfluidity is a second-order phase transition. What continuous symmetry is spontaneously broken? Well, since BEC is the condensation of particles in the ground state, it stands to reason that the symmetry being spontaneously broken is conservation of particle number. In QFT, particle number conservation is implemented by a global $U(1)$ symmetry, so we expect that *superfluidity describes the spontaneous breaking of a global $U(1)$ symmetry*. More precisely, it describes the spontaneous breaking of this symmetry *at finite charge density*, since the superfluid state contains a non-zero number of particles. (Incidentally, if the particles were charged, the $U(1)$ symmetry would be gauged, and its spontaneously broken phase would describe superconductivity.)

The simplest theory of superfluidity at zero temperature is a massive complex scalar field with quartic interactions,

$$\mathcal{L} = -\partial_\mu \psi \partial^\mu \psi - m^2 |\psi|^2 - \frac{g}{2}|\psi|^4. \tag{21}$$

The coupling constant $g$ must be positive to ensure that the potential is bounded below, though we will see that is also required for stability of the superfluid. This theory is invariant under the global $U(1)$ symmetry

$$\psi \to e^{i\alpha}\psi; \qquad \alpha \in \mathbb{R}. \tag{22}$$

The corresponding conserved Noether current is

$$j^\mu = 2\,\text{Im}(\psi^\star \partial^\mu \psi). \tag{23}$$

As usual, its time component gives the charge density, which in this case is the particle number density:

$$n = -2\,\text{Im}(\psi^\star \dot{\psi}). \tag{24}$$

## 3.1 Condensate

To describe the condensate, we work in the mean-field approximation, whereby the condensate wavefunction $\psi_0$ satisfies the classical equation of motion,[3]

$$\partial^2 \psi_0 = m^2 \psi_0 + g|\psi_0|^2 \psi_0. \tag{25}$$

Assuming homogeneity and working in the condensate rest frame, an appropriate ansatz is

$$\psi_0(t) = v e^{i\mu_\text{R} t}, \tag{26}$$

where $\mu_\text{R}$ is the (relativistic) chemical potential. For now you can take this as a definition of the chemical potential, though later on we will provide physical motivation. Substituting this ansatz into (25) gives

$$\mu_\text{R}^2 = m^2 + gv^2. \tag{27}$$

To relate $\mu_\text{R}$ to the non-relativistic chemical potential $\mu$ discussed earlier, take the non-relativistic limit:

$$\mu_\text{R} = \sqrt{m^2 + gv^2} \simeq m + \frac{gv^2}{2m}. \tag{28}$$

The first term is recognized naturally as the rest mass energy, while the second term, which is the energy due to interactions, is identified as the non-relativistic chemical potential:

$$\mu = \frac{gv^2}{2m}. \tag{29}$$

---

[3]As we will see, the interpretation of $\psi_0$ as a wavefunction is justified by the fact $|\psi_0|^2$ gives the mean number density of particles in the mean-field approximation.

Thus $\mu$ is positive (since $g > 0$). This is not inconsistent with our earlier discussion of the free Bose gas. Recall that $\mu \to 0^-$ as $T \to 0$ in that case. Turning on interactions at $T = 0$ then pushes $\mu$ to be slightly positive.

Substituting the ansatz (26) into (24) gives the condensate number density,

$$n = 2\mu_R v^2 \simeq 2mv^2 \,, \tag{30}$$

where in the last step we have taken the non-relativistic limit. Equivalently, using (29) this can be expressed in terms of the chemical potential:

$$n \simeq \frac{4m^2\mu}{g} \,. \tag{31}$$

## 3.2 Phonons

From Goldstone's theorem [54, 55], we expect there should be a massless/gapless mode as a consequence of the spontaneous breaking of the global $U(1)$ symmetry. These gapless excitations are *phonons*. To study them, let us perturb the field as

$$\psi(\vec{x}, t) = (v + h(\vec{x}, t)) e^{i(\mu_R t + \pi(\vec{x}, t))} \,. \tag{32}$$

Substituting into the Lagrangian (21) gives

$$\mathcal{L} = -\left(\partial_\mu h\right)^2 + (v + h)^2 \left[ g v^2 + 2\mu_R \dot{\pi} + \dot{\pi}^2 - \left(\vec{\nabla}\pi\right)^2 \right] - \frac{g}{2}(v + h)^4 \,. \tag{33}$$

Expanding $\mathcal{L}$ to quadratic order, one immediately notices that $h$ has a mass,[4] given by

$$m_h^2 = g v^2 \,. \tag{34}$$

At sufficiently low energy/momentum, we are justified in integrating out $h$, order by order in its derivatives. To zeroth order in this expansion, $h$ reduces to an auxiliary field. In the saddle-point approximation, its equation of motion implies

$$g(v + h)^2 = g v^2 + 2\mu_R \dot{\pi} + \dot{\pi}^2 - \left(\vec{\nabla}\pi\right)^2 \equiv X_R \,. \tag{35}$$

Substituting back into (33) gives

$$\mathcal{L}_\pi = \frac{1}{2g} X_R^2 \,. \tag{36}$$

At this stage it is convenient to take the non-relativistic limit, with $\mu_R \simeq m$ and $\dot{\pi} \ll m$, so that

$$\begin{aligned} X_R &\simeq g v^2 + 2m\dot{\pi} - \left(\vec{\nabla}\pi\right)^2 \\ &= 2m \underbrace{\left( \mu + \dot{\pi} - \frac{\left(\vec{\nabla}\pi\right)^2}{2m} \right)}_{\equiv X} \,, \end{aligned} \tag{37}$$

where in the last step we have used (29). Thus (36) becomes

$$\mathcal{L}_\pi = \frac{2m^2}{g} X^2 \,; \qquad X = \mu + \dot{\pi} - \frac{\left(\vec{\nabla}\pi\right)^2}{2m} \,. \tag{38}$$

---

[4]One should be more careful here. Because of the kinetic mixing term $\dot{\pi}h$, the massive degree of freedom is a linear combination of $\pi$ and $h$, while the gapless mode is the orthogonal linear combination. But you can convince yourself that the leading-order effective Lagrangian (36) ends up being the same.

This is the zero-temperature, non-relativistic effective Lagrangian for the gapless mode $\pi$, to leading order in derivatives. Notice that the original $U(1)$ symmetry (22) acts non-linearly on $\pi$ as a shift symmetry,

$$\pi \to \pi + \alpha. \tag{39}$$

I leave it to you as an exercise to work out the sub-leading corrections to this Lagrangian. To do so, you must keep additional derivative terms in the equation of motion for $h$, and solve this equation perturbatively. The solution for $h$ will be (35) plus terms involving higher-gradients of $\pi$. The resulting effective Lagrangian is of the general form [56]

$$\mathcal{L}_\pi = \frac{2m^2}{g} X^2 + c_1 f_1(X) \left( \vec{\nabla} X \right)^2 + c_2 f_2(X) \left( \vec{\nabla}^2 \pi \right)^2 + \dots \tag{40}$$

To see that the excitations of this field are sound waves or phonons, expand (38) to quadratic order:

$$\mathcal{L}_\pi \simeq \frac{2m^2 \mu^2}{g} + \frac{4m^2 \mu}{g} \dot{\pi} + \frac{2m^2}{g} \left( \dot{\pi}^2 - \frac{\mu}{m} \left( \vec{\nabla} \pi \right)^2 \right). \tag{41}$$

The first term is just a constant and therefore can be dropped. The second term, proportional to $\dot{\pi}$, is a total derivative and can also be dropped. Notice that its coefficient gives the conserved charge, $n = \frac{4m^2 \mu}{g}$, as it should. Lastly, from the remaining terms we see that $\pi$ excitations have a linear dispersion relation $\omega_k = c_s k$, with sound speed

$$c_s^2 = \frac{\mu}{m} = \frac{gn}{4m^3}, \tag{42}$$

where we have used (31).

The linear dispersion relation of phonons leads us to "Landau's criterion": when an impurity moves through a superfluid with subsonic velocity, there is no friction. It is kinematically impossible for an impurity moving subsonically to radiate phonons. If the motion is supersonic, however, phonon radiation becomes possible, analogously to Cerenkov electromagnetic radiation. The superpower of being frictionless originates from the fact that sound waves are the only low-energy excitations in the superfluid.

Two remarks:

- The requirement $g > 0$, which at the level of the original Lagrangian (21) gives a potential that is bounded below, also ensures that the superfluid is stable ($c_s^2 > 0$). Thus stability requires *repulsive* self-interactions.

- Keeping subleading terms in the gradient expansion, as in (40), you will obtain the corrected dispersion relation

$$\omega_k^2 = c_s^2 k^2 + \frac{k^4}{4m^2} + \dots \tag{43}$$

Notice that, as we turn off interactions ($g \to 0$), we have $c_s \to 0$, and the dispersion relation reduces to the standard expression for free, non-relativistic particles: $\omega_k \to \frac{k^2}{2m}$. Thus superfluidity disappears in this limit. In other words, superfluidity requires self-interactions.

## 3.3 Equation of state

Going back to the effective Lagrangian (38), let us set $\pi = 0$ to study the condensate properties. In this case, $X = \mu$, and (38) becomes

$$\mathcal{L} = \frac{2m^2}{g} \mu^2 \equiv P(\mu), \tag{44}$$

where, as usual, we have identified the Lagrangian density with the pressure $P$. This defines the grand canonical equation of state $P(\mu)$.[5] In particular, the standard thermodynamic relation

$$n = \left(\frac{\partial P}{\partial \mu}\right)_T = \frac{4m^2\mu}{g} \tag{45}$$

reproduces our earlier expression (31) for the number density. This is one (roundabout) way to see that $\mu$ is indeed the chemical potential.

## 4 General effective field theory

At this point, we can step back and write down the most general low-energy effective Lagrangian for superfluids at zero temperature, working directly in the non-relativistic regime. (For introductory textbooks on effective field theory, the reader is referred to [57, 58].)

As usual with effective field theory (EFT), the first step is to identify the relevant degree of freedom. At sufficiently low energy/momentum, the relevant field is the Goldstone boson, denoted for now by $\theta$. The next step is to specify the relevant symmetries of the problem. Firstly, we have the $U(1)$ symmetry, which in light of (39) acts non-linearly as a shift symmetry

$$\delta\theta = \alpha. \tag{46}$$

This will be the case if there is at least one derivative per field, that is, $\mathcal{L}_\theta = \mathcal{L}_\theta(\dot\theta, \partial_i\theta, \ldots)$. Secondly, the theory should be invariant under Galilean transformations, which is the relevant symmetry group for non-relativistic physics. Rotational invariance requires spatial gradients to be contracted as $\left(\vec{\nabla}\theta\right)^2$. Less trivial is invariance under Galilean boosts. Recall from quantum mechanics that the phase of the wavefunction (*i.e.*, $\theta$) transforms under Galilean boosts as

$$\delta\theta = m\vec{v}\cdot\vec{x} + t\vec{v}\cdot\vec{\nabla}\theta. \tag{47}$$

Convince yourself that the combination

$$X = \dot\theta - \frac{\left(\vec{\nabla}\theta\right)^2}{2m} \tag{48}$$

transforms as a scalar under (47), that is, $\delta X = t\vec{v}\cdot\vec{\nabla}X$. Therefore, any function of $X$ will be Galilean invariant. Specifically, to leading order in derivatives, the most general Lagrangian compatible with the above symmetries is of the form [59]

$$\mathcal{L}_\theta = P(X). \tag{49}$$

The choice of $P$ specifies the equation of state of the superfluid. Higher-derivative corrections are once again of the form given in (40). As an exercise, check that (49) transforms to a total derivative under (47), hence the action is invariant under Galilean transformations.

It is straightforward to see that the background $\bar\theta(t) = \mu t$ solves the equation of motion. To study phonon excitations around this condensate, let $\theta(\vec{x}, t) = \mu t + \pi(\vec{x}, t)$, such that

$$X = \mu + \dot\pi - \frac{\left(\vec{\nabla}\pi\right)^2}{2m}. \tag{50}$$

I leave it to you as an exercise to check the following facts:

---

[5]In general, the pressure is a function of $\mu$ and $T$, but here we are working at $T = 0$.

i) Setting $\pi = 0$, convince yourself that the charge density satisfies

$$n = \left(\frac{\partial P}{\partial \mu}\right)_T = P_{,X}\big|_{\pi=0} . \tag{51}$$

ii) Expand the Lagrangian to quadratic order in $\pi$, and show that the sound speed is

$$c_s^2 = \frac{1}{m}\frac{P_{,\mu}}{P_{,\mu\mu}} = \frac{P_{,\mu}}{m}\frac{\partial \mu}{\partial n} = \frac{\partial P}{\partial \rho} . \tag{52}$$

This matches the usual expression for the adiabatic sound speed.

Let us mention a few examples of superfluid theories of interest:

- The working example studied in Sec. 3, with $P(X) \sim X^2$, describes particles with $2 \to 2$ interactions. It derives from a complex scalar field with quartic potential, $V(|\psi|) \sim |\psi|^4$. The equation of state, which follows from (31) and (44), is $P(n) \sim n^2$.

- To reproduce the MOND empirical relations in galaxies, the relevant DM superfluid is akin to $P(X) \sim X^{3/2}$. This describes particles with $3 \to 3$ interactions, and can be derived from a complex scalar field with hexic potential, $V(|\psi|) \sim |\psi|^6$. This corresponds to a polytropic equation of state $P(n) \sim n^3$. Thus, despite the non-analytic dependence on $X$, the equation of state is perfectly analytic in $n$.

- A well-known example of a theory with fractional power in cold atom systems is the Unitary Fermi Gas (UFG) [60, 61], which describes fermionic atoms tuned at unitary. The UFG superfluid action is fixed by non-relativistic scale invariance to the non-analytic form $\mathcal{L}_{\mathrm{UFG}}(X) \sim X^{5/2}$ [56].

The effective theory (49) can easily be generalized to include the interaction energy $V(\vec{x})$ associated with an external potential. For cold atoms in the laboratory, this describes the trapping potential. For the case of interest of DM in galaxies, this describes the gravitational potential energy $V(\vec{x}) = m\Phi(\vec{x})$, where $\Phi$ is the Newtonian potential. From the perspective of phonons, this amounts to a shift in the chemical potential. In other words, (49) still applies, with $X$ now given by[6]

$$X = \mu - m\Phi(\vec{x}) + \dot{\pi} - \frac{(\vec{\nabla}\pi)^2}{2m} . \tag{53}$$

Thus, the Lagrangian describing a DM superfluid coupled to Newtonian gravity is

$$\mathcal{L} = -\frac{1}{8\pi G_N}\left(\vec{\nabla}\Phi\right)^2 + P(X) . \tag{54}$$

As a consistency check, varying with respect to $\Phi$ and ignoring phonons ($\pi = 0$), we obtain Poisson's equation,

$$\vec{\nabla}^2\Phi = 4\pi G_N m P_{,X}\big|_{\pi=0} = 4\pi G_N \rho , \tag{55}$$

where we have used (51).

For completeness, since the DM condensate in actual galactic halos has a velocity dispersion, we should discuss the generalization of the superfluid EFT for non-zero temperature. At finite sub-critical temperature, a superfluid is described phenomenologically by Landau's

---

[6]To convince yourself, go back to the quartic theory (21) and add the coupling to gravity $\mathcal{L}_{\mathrm{grav}} = -\Phi m|\psi|^2$. Upon integrating out $h$, you will find that $X$ now takes the form (53).

two-fluid model: an admixture of a superfluid component and a normal component. The finite-temperature effective Lagrangian is a function of three scalars [62]:

$$\mathcal{L}_{T \neq 0} = F(X, B, Y). \tag{56}$$

The scalar $X$ describes phonon excitations, as before. The remaining scalars are defined in terms of the three Lagrangian coordinates $\psi^I(\vec{x}, t)$, $I = 1, 2, 3$ of the normal fluid:

$$
\begin{aligned}
B &\equiv \sqrt{\det \partial_\mu \psi^I \partial^\mu \psi^J}\,; \\
Y &\equiv u^\mu \left( \partial_\mu \theta + m \delta_\mu^0 \right) - m \simeq \mu - m\Phi + \dot{\pi} + \vec{v} \cdot \vec{\nabla} \pi\,,
\end{aligned} \tag{57}
$$

where $u^\mu = \frac{1}{6\sqrt{B}} \epsilon^{\mu\alpha\beta\gamma} \epsilon_{IJK} \partial_\alpha \psi^I \partial_\beta \psi^J \partial_\gamma \psi^K$ is the unit 4-velocity vector, and in the last step for $Y$ we have taken the non-relativistic limit $u^\mu \simeq (1 - \Phi, \vec{v})$. By construction, these scalars respect the internal symmetries: i) $\psi^I \to \psi^I + c^I$ (translations); ii) $\psi^I \to R^I_J \psi^J$ (rotations); iii) $\psi^I \to \xi^I(\psi)$, with $\det \frac{\partial \xi^I}{\partial \psi^J} = 1$ (volume-preserving reparametrizations).

## 5 MOND phenomenology from DM superfluidity

Once we take seriously the idea that DM is in a superfluid phase inside galaxies, the key question is — what kind of superfluid? Motivated by the empirical galactic scaling relations, in [32,33] we conjectured that DM phonons are described by the non-relativistic MOND scalar action,[7]

$$P(X) = \frac{2\Lambda(2m)^{3/2}}{3} X \sqrt{|X|}. \tag{58}$$

As mentioned earlier, the fractional power of the kinetic term would be strange if (58) described a fundamental scalar field. As a theory of phonons, however, the power determines the superfluid equation of state, and fractional powers are not uncommon. For instance, the effective theory for the UFG superfluid is also non-analytic.

To mediate a MONDian force between ordinary matter, phonons must couple to baryons through

$$\mathcal{L}_{\text{int}} \sim \frac{\Lambda}{M_{\text{Pl}}} \pi \rho_{\text{b}}\,, \tag{59}$$

where $\rho_{\text{b}}$ is the baryon mass density, and $M_{\text{Pl}} = \frac{1}{\sqrt{8\pi G_N}}$ is the reduced Planck mass. This operator explicitly breaks the shift symmetry (39), but the breaking is $M_{\text{Pl}}$-suppressed and thus technically natural from an EFT point of view. It could arise, for instance, if the superfluid includes two components coupled through a Rabi-Josephson interaction [64]. With the field redefinition $\phi = \frac{\Lambda}{M_{\text{Pl}}} \pi$, we see that this theory reproduces the MOND Lagrangian (3) for

$$\Lambda \sim \sqrt{a_0 M_{\text{Pl}}} \sim \text{meV}. \tag{60}$$

If we recall from Sec. 2 that the DM mass can be at most $m \sim \text{eV}$ in order to have BEC in galaxies, we see that all scales in the problem are nicely eVish.

With this action, the DM superfluid gives rise to a long-range, phonon-mediated force between ordinary matter particles:

$$a_\pi = \sqrt{a_0 a_{\text{b}}}\,, \tag{61}$$

---

[7]The square-root form ensures that the action is well-defined for time-like field profiles, and that the Hamiltonian is bounded below [63].

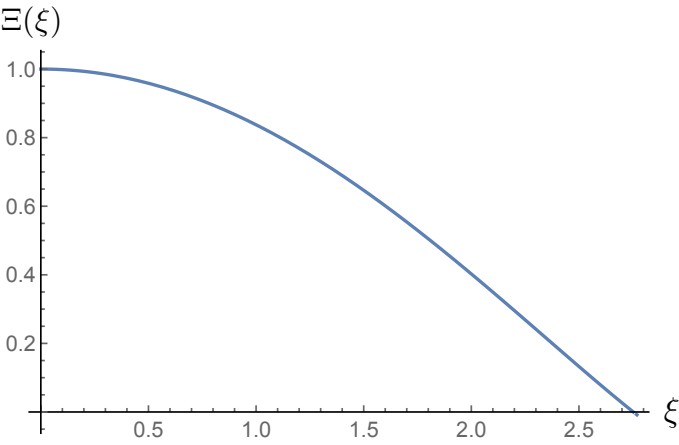

Figure 1: Numerical solution of Lane-Emden equation (67).

where $a_b$ is the Newtonian acceleration due to baryons only. Unlike "pure" MOND, however, the DM halo itself contributes a gravitational acceleration $a_{DM}$. This contribution is negligible on distances probed by galactic rotation curves, but becomes comparable to the MOND component at distances of order the size of the superfluid core. In other words, the total acceleration acting on baryons is

$$\vec{a} = \vec{a}_b + \vec{a}_{DM} + \vec{a}_\pi \,. \tag{62}$$

Below we will derive the density profile of the superfluid core at zero temperature.

It is important to stress that, unlike most attempts to modify gravity, there is no fundamental additional long-range force in the model. Instead the phonon-mediated force is an *emergent* phenomenon which requires the coherence of the underlying superfluid substrate. This has important implications for the phenomenological viability of the scenario *vis-à-vis* solar system tests of gravity. Although typical accelerations in the solar system are large compared to $a_0$, post-Newtonian tests are sensitive to small corrections to Newtonian gravity and require further suppression of the new force at short scales. The superfluid framework offers an elegant explanation. As argued in [32, 33], the large phonon gradient induced by an individual star results in a breakdown of superfluidity in its vicinity. Coherence is lost within the solar system. Individual DM particles still interact with baryons, but no long-range force can be mediated.

Applying the results of the previous Sections, you can check that the condensate equation of state is

$$P(\mu) = \frac{2\Lambda}{3}(2m\mu)^{3/2} \,. \tag{63}$$

The number density of condensed particles is $n = \Lambda(2m)^{3/2}\mu^{1/2}$, thus the equation of state can be expressed equivalently as

$$P = \frac{\rho^3}{12\Lambda^2 m^6} \,. \tag{64}$$

Lastly, phonons propagate with sound speed:

$$c_s^2 = \frac{2\mu}{m} = \frac{\rho^2}{4\Lambda^2 m^6} \,. \tag{65}$$

To gain intuition, it is instructive to study the density profile of a spherically-symmetric DM superfluid core, ignoring baryons. The condition of hydrostatic equilibrium implies

$$\frac{1}{\rho(r)}\frac{dP(r)}{dr} = -\frac{4\pi G_N}{r^2}\int_0^r dr' r'^2 \rho(r') \,. \tag{66}$$

Substituting the equation of state (64), and introducing the dimensionless variables $\rho = \rho_{\text{core}} \Xi^{1/2}$ and $r = \sqrt{\frac{\rho_{\text{core}}}{32\pi G_{\text{N}} \Lambda^2 m^6}} \xi$, with $\rho_{\text{core}}$ denoting the central density, you can show that (66) becomes

$$\left( \xi^2 \Xi' \right)' = -\xi^2 \Xi^{1/2} \,, \tag{67}$$

where $' \equiv \mathrm{d}/\mathrm{d}\xi$. This is a Lane-Emden equation. The numerical solution, with boundary conditions $\Xi(0) = 1$ and $\Xi'(0) = 0$, is shown in Fig. 1. We see that the resulting superfluid density profile is cored. The density is found to vanish at $\xi_1 \simeq 2.75$, which defines the core radius: $R_{\text{core}} = \sqrt{\frac{\rho_{\text{core}}}{32\pi G_{\text{N}} \Lambda^2 m^6}} \xi_1$.

Meanwhile the central density is related to the core mass as [65] $\rho_{\text{core}} = \frac{3 M_{\text{core}}}{4\pi R_{\text{core}}^3} \frac{\xi_1}{|\Xi'(\xi_1)|}$, with $\Xi'(\xi_1) \simeq -0.5$. Combining these results, it is straightforward to solve for the core radius:

$$R_{\text{core}} \simeq \left( \frac{M_{\text{core}}}{10^{11} \mathrm{M}_\odot} \right)^{1/5} \left( \frac{m}{\mathrm{eV}} \right)^{-6/5} \left( \frac{\Lambda}{\mathrm{meV}} \right)^{-2/5} 45 \text{ kpc} \,. \tag{68}$$

Remarkably, for $m \sim \mathrm{eV}$ and $\Lambda \sim \mathrm{meV}$ we obtain DM cores of realistic size!

The above calculation, while instructive, ignores a number of effects which are important in deriving realistic density profiles in galaxies. Firstly, the addition of baryons results in a phonon gradient, which is not necessarily small compared to the chemical potential. Indeed, the deep-MOND acceleration law ($a \simeq \sqrt{a_{\text{b}} a_0}$) is recovered whenever the phonon gradient dominates over the chemical potential, $(\vec{\nabla}\pi)^2 \gg \mu m$. The story is further complicated further by the fact that perturbations around this zero-temperature, static background are unstable (ghost-like). However this instability can naturally be cured by finite-temperature effects [32, 33]. Indeed, owing to their velocity dispersion DM particles have a small non-zero temperature in galaxies, hence we expect the zero-temperature Lagrangian (58) to receive finite-temperature corrections in galaxies. The finite-temperature equation of state for DM superfluids was calculated in [66].

In [35] we calculated an approximate finite-temperature density profile, consisting of a superfluid core, with approximately homogeneous density, surrounded by an envelope of normal-phase DM particles following an NFW profile. We explicitly fitted this density profile to two representative galaxies: a representative low-surface brightness galaxy IC 2574, and a representative high-surface brightness galaxy UGC 2953. See also [67]. The superfluid model offers an excellent fit in both cases. See [35, 68–70] for phenomenological implications of DM superfluidity for other astrophysical systems.

A distinctive prediction of DM superfluidity is the absence of dynamical friction for subsonic motion within the superfluid region [48, 70, 71]. This may alleviate a number of minor problems for $\Lambda$CDM. For instance, instead of being slowed down by dynamical friction, galactic bars in spiral galaxies should achieve a nearly constant velocity [72], as favored by observations [73].

It may also offer a natural explanation to the long-standing puzzle of why the five globular clusters orbiting Fornax have not merged to the center to form a stellar nucleus. Indeed, in the context of CDM, dynamical friction should have caused the globular clusters to rapidly fall towards the center of Fornax [74, 75]. In reality Fornax shows no sign of such mergers. See [76, 77] for possible explanations within the classical DM particles context. In superfluid DM, the globular clusters are happily swimming within the superfluid core without dissipation.

A potential key difference with $\Lambda$CDM is the merger rate of galaxies. If the infall velocity is subsonic, then halos will pass through each other with negligible dissipation, possibly resulting in multiple encounters and a longer merger time. If the infall velocity is supersonic, however, the encounter will excite DM particles out of the condensate, resulting in dynamical friction and standard merger dynamics.

# 6 Other descriptions

For completeness let us briefly describe other well-studied theoretical descriptions of superfluidity. For this purpose we will focus on the quartic theory (21) for concreteness. Our starting point is to take the non-relativistic limit directly at the level of the action via the field redefinition

$$\psi(\vec{x},t) = \frac{1}{\sqrt{2m}}\Psi(\vec{x},t)e^{-imt}\,. \tag{69}$$

Substituting into (21) and neglecting terms quadratic in time derivatives, we obtain

$$\mathcal{L} = \frac{i}{2}\left(\Psi^{\star}\dot{\Psi} - \Psi\dot{\Psi}^{\star}\right) - \frac{1}{2m}\left|\vec{\nabla}\Psi\right|^{2} - \frac{g}{8m^{2}}|\Psi|^{4}\,. \tag{70}$$

Varying with respect to $\Psi^{\star}$ gives

$$i\dot{\Psi} = \left(-\frac{1}{2m}\vec{\nabla}^{2} + \frac{g}{4m^{2}}|\Psi|^{2}\right)\Psi\,. \tag{71}$$

This non-linear Schrödinger equation is the celebrated Gross–Pitaevskii equation [78, 79].

Performing the Madelung decomposition, familiar from ordinary quantum mechanics [80],

$$\Psi(\vec{x},t) = \sqrt{\frac{\rho(\vec{x},t)}{m}}e^{i\theta(\vec{x},t)}\,, \tag{72}$$

and defining the superfluid velocity

$$\vec{v}(\vec{x},t) = \frac{1}{m}\vec{\nabla}\theta\,, \tag{73}$$

the Gross–Pitaevskii equation gives

$$\begin{aligned}
\dot{\rho} + \vec{\nabla}\cdot(\rho\vec{v}) &= 0\,; \\
\rho\left(\dot{\vec{v}} + \left(\vec{v}\cdot\vec{\nabla}\right)\vec{v}\right) &= -\vec{\nabla}P + \frac{n}{2m^{2}}\frac{\vec{\nabla}^{2}\sqrt{n}}{\sqrt{n}}\,,
\end{aligned} \tag{74}$$

where $P = \frac{g}{8m^{4}}\rho^{2}$ is the pressure. These are the well-known equations of hydrodynamics. The first equation is the continuity equation, while the second is Euler's equation. Since the latter is dissipation-free, the flow is inviscid. The last term in Euler's equation, $\sim \frac{\vec{\nabla}^{2}\sqrt{n}}{\sqrt{n}}$, is the so-called "quantum pressure" term, which plays a key role in stabilizing the cored profiles in fuzzy DM [38, 48]. In our case, this is subdominant relative to the classical pressure term $-\vec{\nabla}P$. Furthermore, since the velocity is the gradient of a scalar, per (73), the flow is irrotational.

# 7 Conclusion

In these lectures we discussed a novel theory of DM superfluidity that reconciles the stunning success of MOND on galactic scales with the triumph of the $\Lambda$CDM model on cosmological scales. The theory of DM superfluidity has some commonalities with self-interacting DM and fuzzy DM. All three proposals achieve a cored DM profile in central regions of galaxies, either through interactions or quantum effects, to alleviate existing observational puzzles/tensions with $\Lambda$CDM on galactic scales. In all three proposals, the DM outside the core is in approximately collisionless form and assumes an NFW profile. Like the quantum pressure of fuzzy DM, the classical pressure of superfluid DM results in lower central densities and implies a minimal halo mass necessary for collapse and virialization.

The main difference is that DM superfluidity achieves a much larger core, encompassing the entire range of scales probed by rotation curve observations. Nevertheless, the superfluid core makes up only a modest fraction of the entire halo. It is large enough to encompass the observed rotation curves, since the phonon force is critical for reproducing MOND. But it is small enough that most of the mass lies in the approximately collisionless envelope, resulting in triaxial halos near the virial radius. The superfluid collective excitations (phonons) mediate a long-range force within the core, thereby affecting the dynamics of orbiting baryons and reproducing the MOND phenomenology.

## Acknowledgements

I wish to warmly thank Marco Cirelli, Babette Dobrich and Jure Zupan for organizing, against all odds, an excellent Les Houches summer school. I also want to thank all the students at the School for their enthusiasm and many stimulating questions. Many thanks to Alexis Bourdon for carefully going through these notes and pointing out some typos. We are grateful to the anonymous referees for their constructive suggestions. Last but not least, I want to thank all my collaborators on this work, particularly Lasha Berezhiani.

**Funding information**    This work was supported in part by the US Department of Energy (HEP) Award DE-SC0013528, NASA ATP grant NNH17ZDA001N, the Charles E. Kaufman Foundation of the Pittsburgh Foundation, and a W. M. Keck Foundation Science and Engineering Grant.

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
