# Peer review of "Dark Matter Superfluidity"

_SciPost Physics Lecture Notes, doi:SciPost Phys. Lect. Notes 42 (2022)_

## Round 1 · Referee Report · Anonymous · 2021-10-18

Strengths

1. This article gives an introduction to one of the most interesting dark matter models to have been introduced in recent years.

2. It is at a level more accessible to students than anything that can be found in the existing literature.

Weaknesses

This is not really a weakness, only a modest wish from someone who will give this reference to some of his students, based on their typical background in his corner of the world:

1. Bose-Einstein condensation, which most students will have been introduced to in a course on statistical physics, is presented in quite some detail. In contrast, some knowledge of effective field theory, which students with only a basic course in quantum field theory will not have been exposed to, is more or less taken for granted.

Report

The acceptance criteria are clearly met: The article provides a correct, very systematic and intelligible presentation of a topic of ongoing interest. It will be a very valuable reference for students (and their thesis advisors).

It may be an oversight of my part, but I cannot see that [68]-[70] in the reference list are referred to in the text. It would perhaps be natural to say something about vortices after concluding in section 6 that the flow is irrotational, and in that case these references would be relevant.

Requested changes

1. Add a reference to an introductory text on effective field theory at the start of section 4.

  • validity: top
  • significance: top
  • originality: top
  • clarity: top
  • formatting: excellent
  • grammar: perfect

Author:  Justin Khoury  on 2021-11-06  [id 1915]

(in reply to Report 2 on 2021-10-18)

We thank the referee for his/her thorough reading of the manuscript and constructive comments. We respond below to his/her specific suggestion.

  1. Add a reference to an introductory text on effective field theory at the start of section 4.

We have added a reference to two recent introductory textbooks on effective field theory.

---

## Round 1 · Referee Report · Anonymous · 2021-10-19

Report

In these lectures, the author introduces and motivates the hybrid DM-MOND model superfluid dark matter. I find them to be clear and well-written, providing a good basis to understand superfluid dark matter.

I have a few comments and suggestions:

1. On p. 2, references [2, 3] are not really about the MDAR, as suggested in the text, but about the RAR. Of course the RAR and the MDAR are equivalent, but giving only these references might confuse readers who don't already know this.

2. On p.3, below (4), the author writes that there is a solution $a_\phi = \sqrt{a_b a_0}$. This is usually a good approximation but is strictly true only in spherical symmetry and other special cases. It might be good to note this to avoid confusion for readers who try to rederive this result.

3. On p. 4, I think the exponent in (6) is missing an overall minus sign. Similarly, I think the right-hand side of (8) on p. 5 is missing an overall minus sign.

4. On p. 8, $\psi_0$ is introduced as the condensate wavefunction. I know it is not uncommon to refer to $\psi$ as a wavefunction, but I find this causes a lot of confusion for students since $\psi$ is not actually a wave function, $|\psi|^2$ is not a probability density. Conceptually, $\psi$ is much closer to the quantum/classical electromagnetic field (or its four-potential) than to a QM wavefunction. It would be good to change the naming or briefly explain this to avoid confusion.

5. On p. 14, the author derives the phonon force and the density profile from the Lagrangian (58). But opposite limits are used for the phonon force and the density profile. For the phonon force to look like $\sqrt{a_0 a_b}$ (eq. (61)), the gradient terms (with $\vec{\nabla} \theta$ or $\vec{\nabla} \pi$) in the Lagrangian must dominate the chemical potential terms (with $\mu$). For the density profile to have, e.g., the equation of state claimed in (63) / (64) the opposite limit is needed. Maybe the density profile (but not the phonon force) is supposed to be for the case without any baryons? In this case it should probably be pointed out that the derived density profile does not directly apply to galaxies. In any case, the two opposite assumptions about which terms dominate should be made explicit to avoid confusion.

6. On p. 15, the author writes that it is not surprising that the density profile is cored. I'm not quite sure what this refers to. Is it that Lane-Emden equations generally give cored profiles? A clarification would be helpful.

  • validity: -
  • significance: -
  • originality: -
  • clarity: -
  • formatting: -
  • grammar: -

Author:  Justin Khoury  on 2021-11-06  [id 1914]

(in reply to Report 1 on 2021-10-19)

We thank the referee for his/her thorough reading of the manuscript and constructive suggestions. We respond below to their specific comments.

  1. On p. 2, references [2, 3] are not really about the MDAR, as suggested in the text, but about the RAR. Of course the RAR and the MDAR are equivalent, but giving only these references might confuse readers who don't already know this.

The referee is correct. To be precise, we have changed the terminology from MDAR to RAR in the revised manuscript.

  1. On p.3, below (4), the author writes that there is a solution a_\phi = \sqrt{a_b a_0}. This is usually a good approximation but is strictly true only in spherical symmetry and other special cases. It might be good to note this to avoid confusion for readers who try to rederive this result.

We have specified that the solution ignores a homogeneous curl term, which vanishes in particular for spherical symmetry.

  1. On p. 4, I think the exponent in (6) is missing an overall minus sign. Similarly, I think the right-hand side of (8) on p. 5 is missing an overall minus sign.

We have corrected these typos.

  1. On p. 8, \psi_0 is introduced as the condensate wavefunction. I know it is not uncommon to refer to \psi as a wavefunction, but I find this causes a lot of confusion for students since \psi is not actually a wave function, |\psi|^2 is not a probability density. Conceptually, \psi is much closer to the quantum/classical electromagnetic field (or its four-potential) than to a QM wavefunction. It would be good to change the naming or briefly explain this to avoid confusion.

In mean field, \psi_0 is commonly referred to as the condensate wavefunction. Because this is standard terminology, I would prefer to keep it as such. The referee’s point is well-taken, so to clarify things I have added a footnote that |\psi_0|^2 gives the average number density in mean field.

  1. On p. 14, the author derives the phonon force and the density profile from the Lagrangian (58). But opposite limits are used for the phonon force and the density profile. For the phonon force to look like \sqrt{a_b a_0}(eq. (61)), the gradient terms in the Lagrangian must dominate the chemical potential terms. For the density profile to have, e.g., the equation of state claimed in (63) / (64) the opposite limit is needed. Maybe the density profile (but not the phonon force) is supposed to be for the case without any baryons? In this case it should probably be pointed out that the derived density profile does not directly apply to galaxies. In any case, the two opposite assumptions about which terms dominate should be made explicit to avoid confusion.

The referee is of course correct, and we have added some clarifying remarks in the paragraph below Eq. (68).

  1. On p. 15, the author writes that it is not surprising that the density profile is cored. I'm not quite sure what this refers to. Is it that Lane-Emden equations generally give cored profiles? A clarification would be helpful.

The Lane-Emden equation for polytropes indeed yield cored profiles, but this is a consequence of the assumed boundary condition at r= 0. The “not surprising” qualifier was unnecessary, and I have decided to remove it to avoid confusion.

---

## Round 1 · Referee Report · Elisa Ferreira · 2021-11-14

Strengths

1. Clear review of the status of the field of DM, introducing one of the best overviews of its states to date
2. Extremely good review of the necessary topics to understand the model
3. Very clear, pedagogic and informative; it is dense (and many times I wish he could expand on these topics given his clear way of explaining) but still very clear

Weaknesses

None

Report

I believe this review represents a well-written, high quality and highly necessary text on the new and exciting model of dark matter, the DM superfluid model. There are other texts from the author and reviews from other authors about the topic, but none of them is as good as this review. In this text, the author presents a very good review on the state of the dark matter field today that although short is so informative that I believe all researchers and students in the field of dark matter and cosmology should read. The text is dense and full of information but that the author explains with clarity, and adding his point of view that I believe is very modern and presents the most fair and complete view of the field.
The author then proceeds to the part that I enjoy the most of the review which is a review on the topics of BEC, superfluidity, QFT and symmetry breaking. This review of basic topics is conducted with absolute clarity and it is highly informative. The author then proceeds to reviewing the DM model that he invented together with his collaborator.

I truly believe that although small this review is extremely well written and very useful fro students and researchers in the field. I will recommend it to my students and collaborators.

I just wish Prof. Khoury is invited to write a longer review on this topic where he can expand more in these topics using his clarity, pedagogics and expertise in the field.

Therefore, I recommend this review to be published as it is. I just suggest one addition as it can be seen below.

Requested changes

In my view, the review can be publish as it is now. The only minor change that I would ask is that the author inform in some place in the review that “natural units” are being used. An attentive student would miss the factors of \hbar and c in the review, so I think a note saying that natural units are adopted would be appropriate.

I recommend the review to be published and don’t need to see the new changes

  • validity: top
  • significance: top
  • originality: high
  • clarity: top
  • formatting: excellent
  • grammar: perfect

Author:  Justin Khoury  on 2022-01-30  [id 2130]

(in reply to Report 4 by Elisa Ferreira on 2021-11-14)

We thank Dr. Ferreira for her thoughtful and positive response. We have added a footnote at the onset of Sec. 2 to alert the reader about natural units.

---

## Round 1 · Referee Report · Anonymous · 2021-11-14

Report

This article discusses the main motivation and formulation behind the development of a very interesting dark matter model known as superfluid dark matter. It is a very well written comprehensive introduction to dark matter superfluidity. This article satisfies all the acceptance criteria and would be very beneficial to students.
There are only few minor suggestions/changes from my part:
1) The line below Eq. (7) contains an extra 'the'.
2) In the footnote on page 6, it will be nice if a reference corresponding to cold atom is also given.
3) In Section 3., line 5, please correct the sentence 'We will primarily interested ...'.
4) In section 3.2, it will be nice if a reference corresponding to Goldstone's theorem is given.
5) Correct the last line on page 16 - 'of a scalar, per(73)...'.

  • validity: -
  • significance: -
  • originality: -
  • clarity: -
  • formatting: -
  • grammar: -

Author:  Justin Khoury  on 2022-01-30  [id 2131]

(in reply to Report 3 on 2021-11-14)

We thank the referee for his/her thorough reading of the manuscript and constructive suggestions. See below for detailed changes.

1) The line below Eq. (7) contains an extra 'the'.

We have corrected this typo.

2) In the footnote on page 6, it will be nice if a reference corresponding to cold atom is also given.

We have added a reference to a classic paper in the field.

3) In Section 3., line 5, please correct the sentence 'We will primarily interested ...'.

Typo corrected.

4) In section 3.2, it will be nice if a reference corresponding to Goldstone's theorem is given.

We have added the appropriate references.

5) Correct the last line on page 16 - 'of a scalar, per(73)...'.

We believe that sentence is correct.

---

## Editorial Decision

published